# MCMC Methods for Parameter Estimation in ODE Systems for CAR-T Cell Cancer Therapy

**DOI:** 10.3390/cancers16183132

**Published:** 2024-09-11

**Authors:** Elia Antonini, Gang Mu, Sara Sansaloni-Pastor, Vishal Varma, Ryme Kabak

**Affiliations:** 1Cilag GmbH International, 6300 Zug, Switzerland; gmu6@its.jnj.com; 2Actelion Pharmaceuticals Ltd., 4123 Allschwil, Switzerland; ssansalo@its.jnj.com; 3Johnson & Johnson World Headqtrs US, Bridgewater, NJ 08807, USA

**Keywords:** ODEs, MCMC, CAR-T cell therapy, immunotherapy, oncology, Metropolis–Hastings, differential evolution, Python, parameter estimation, Bayesian, Monte Carlo, Markov chain

## Abstract

**Simple Summary:**

Chimeric antigen receptor (CAR)-T cell therapy is a promising treatment for highly resistant blood cancers, using genetically modified T cells from the patient or a donor. While CAR-T therapy has been successful in pre-clinical and clinical stages for cancer treatment, it also presents challenges, including cytokine release syndrome. To improve the efficacy and reduce side effects, there is a need to better understand CAR-T cell behavior. We aimed to develop a mathematical framework that describes CAR-T behavior using ordinary differential equations (ODEs) and Bayesian parameter estimation (using advanced algorithms including Metropolis–Hastings, DEMetropolis, and DEMetropolisZ). This model will help to understand CAR-T behavior and, by extension, help to improve the effectiveness and efficacy of therapy in a clinical setting.

**Abstract:**

Chimeric antigen receptor (CAR)-T cell therapy represents a breakthrough in treating resistant hematologic cancers. It is based on genetically modifying T cells transferred from the patient or a donor. Although its implementation has increased over the last few years, CAR-T has many challenges to be addressed, for instance, the associated severe toxicities, such as cytokine release syndrome. To model CAR-T cell dynamics, focusing on their proliferation and cytotoxic activity, we developed a mathematical framework using ordinary differential equations (ODEs) with Bayesian parameter estimation. Bayesian statistics were used to estimate model parameters through Monte Carlo integration, Bayesian inference, and Markov chain Monte Carlo (MCMC) methods. This paper explores MCMC methods, including the Metropolis–Hastings algorithm and DEMetropolis and DEMetropolisZ algorithms, which integrate differential evolution to enhance convergence rates. The theoretical findings and algorithms were validated using Python and Jupyter Notebooks. A real medical dataset of CAR-T cell therapy was analyzed, employing optimization algorithms to fit the mathematical model to the data, with the PyMC library facilitating Bayesian analysis. The results demonstrated that our model accurately captured the key dynamics of CAR-T cell therapy. This conclusion underscores the potential of parameter estimation to improve the understanding and effectiveness of CAR-T cell therapy in clinical settings.

## 1. Introduction

CAR-T is based on the genetic modification of T cells (from the patient or a donor) to recognize antigens present on the targeted tumor cell’s surface [1]. Over the years, a new generation of CAR constructs have incorporated co-stimulatory domains, to increase their anti-tumor activity [2].

CAR-T therapy has become a lifesaving treatment for many oncology patients. The results from different studies show that the ORR (overall response rate) is 44–91% and CR (complete remission) rates are 28–68% [3]. However, CAR-T cells are associated with acute severe events, such as CRS, aplasia, cytopenias, and infections, among others. Therefore, there is a medical need to better understand the complex mechanisms behind these events.

The effectiveness of CAR-T cell therapy largely depends on the patient’s T cell quality (cell proliferation, life time, and anti-tumor activity). Therefore, a key goal for better outcomes is to prevent or delay the T cells from becoming exhausted ($C_{E}$), while keeping their memory properties ($C_{M}$). Exhausted CAR-T cells are a limitation to the therapy efficiency and are defined as inhibition of the T cell proliferation and effector function. This exhaustion happens due to different factors, such as persistent antigen stimulation, the immunosuppressive tumor environment [4], and in vitro growing system [5], and it leads to resistance and relapse in patients. Currently, different strategies are being studied to overcome exhaustion in CAR-T cell therapy. For instance, immune checkpoint blockade (PD-1) [6,7] or/and TGF-β [8], deletion of *Cbl-b* [9], modulation of CAR surface expression [10], and uncoupling of antigen recognition [11]. Furthermore, studies with patients have suggested that having more memory-like T cells leads to a more efficient treatment, meaning CAR-T cells work for longer, with a slower disease progression and longer remission periods for patients. Therefore, analyzing the different types of T cells in the final treatment product and in patient samples after treatment could help to better understand how different immune characteristics can make CAR-T cell therapy more effective. Figure 1 describes the interactions between the CAR-T phenotypes and tumor cells (T).

In recent years, various ordinary differential equation (ODE) models have been developed to understand the dynamics of CAR-T cell kinetics and their interactions with cancer cells. A range of studies have reviewed the CAR-T structure, pharmacological properties, and modeling approaches, highlighting unique challenges due to CAR-T cells’ live nature and their atypical kinetic profiles compared to traditional drugs [12,13,14,15,16]. Among these models, the ODE model from [17] was selected for its ability to capture the kinetics of CAR-T cells, including antigen-dependent expansion, phenotypic heterogeneity, and the influences of the tumor. This model uniquely integrates functional, memory, and exhausted CAR-T cell phenotypes with cancer cell dynamics, providing a comprehensive description of CAR-T immunotherapy’s effectiveness in patients with hematological cancers. By fitting this model to clinical data, this allows for the identification of key kinetic parameters and phenotypic dynamics that correlate with long-term therapy outcomes.
Figure 1Different CAR-T cell phenotypes and their effect on patients, every dynamic is labeled with the kinetic parameters from Table 1 and Table 2. After CAR-T cell infusion, they spread throughout the body (CD), with some settling in blood and tumor areas. These engrafted cells are known as effector. CAR-T cells (CT), they expand upon antigen contact (**—**,**—**) and differentiate into memory CAR-T cells (CM), before becoming exhausted and either dying naturally or through a tumor immunosuppressive mechanism. Both effector CAR-T cells and those that are rapidly spreading can kill cancer cells. The memory CAR-T cells eventually die off, yet they can quickly react to cancer cells. When these memory cells come into contact with cancer cells, they turn back into effector CAR-T cells and quickly help to fight the cancer. However, over time, these cells can become worn out and stop working. The cancer cells (T), which are recognized by the CAR-T cells, grow based on what they have available in their environment and are killed off by the cytotoxic effect of the working CAR-T cells (**—**). The net growth in cancer cells is determined by how fast they multiply and naturally they die, represented by the logistic function (**—**) [17].
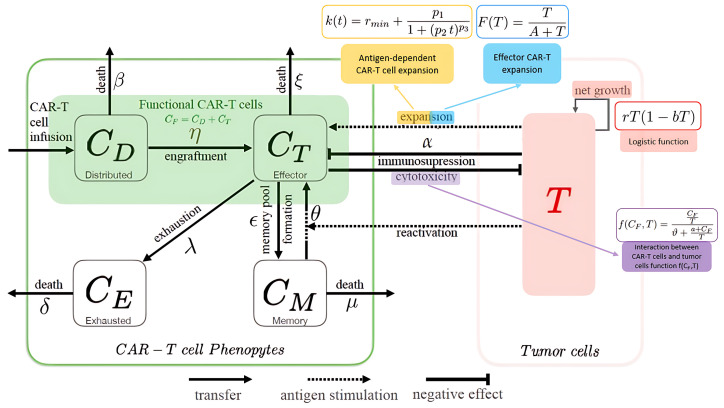


An important tool used in this paper are Markov chains [18], which form the basis of many stochastic models in this study. This serves as a foundation for developing advanced Markov models and Monte Carlo simulation techniques, which are crucial for capturing the probabilistic nature of biological interactions and treatment responses.

Furthermore, this paper delves into Bayesian statistics to estimate the parameters of our mathematical model. Techniques such as Monte Carlo integration [19], Bayesian inference, and Markov chain Monte Carlo (MCMC) methods are employed to assess the uncertainty in parameter estimates and enhance the model’s predictive capabilities. Recent advancements in Bayesian methods have provided powerful tools for biological parameter estimation [20,21], particularly in ODE models, such as that of Keersmaekers et al. for modeling the B- and T-cell dynamics induced by Varicella-Zoster virus vaccines [22], or in the novel approach presented by Dang et al., where conditional neural ODE processes are utilized for forecasting individual disease progression, as demonstrated in a case study on COVID-19 [23].

MCMC methods, in particular, are crucial to the understanding and implementation of Bayesian parameter estimation within our context. This paper provides an in-depth examination of several MCMC techniques, including the standard Metropolis–Hastings algorithm. Additionally, we explore advanced methods like the DEMetropolis [24] and DEMetropolisZ algorithms [25], which integrate differential evolution [26] with the classical Metropolis–Hastings method, to enhance convergence rates. Recent research by Valderrama-Bahamóndez et al. demonstrated the applicability of these methods to ODE biological models, further validating their use in our analysis [27].

The theoretical findings and algorithms proposed in this paper were substantiated through a numerical application using Python, with Jupyter Notebooks serving as the computational platform. A real medical dataset [15] of CAR-T cell therapy was analyzed, and various optimization algorithms were employed to fit the mathematical model to the data. The PyMC library [28] was utilized to construct MCMC samplers, facilitating the Bayesian analysis.

This paper aimed to not only study the dynamics of CAR-T cell interactions within a patient’s body but also to explore the frontier of mathematical methodologies that can be used in medical research to create a patient-specific treatment [17].

## 2. Methods

The parameters were extracted from the mathematical model of A. Paixao. et al. [17]. Different phenotypes of CAR-T cells were considered:*Distributed* functional CAR-T cell ($C_{D}$);*Effector* functional CAR-T cell ($C_{T}$);*Memory* CAR-T cell ($C_{M}$);*Exhausted* CAR-T cell ($C_{E}$).

In the monitoring phase, only the total number of CAR-T cells (*C*) can be measured
(1)$$C=C_{D}+C_{T}+C_{M}+C_{E}.$$

### 2.1. Math Model Description

The model from [17] describes CAR-T cell immunotherapy in blood cancer patients, including distributed $C_{D}$ and functional $C_{T}$, memory $C_{M}$, and exhausted CAR-T cells $C_{E}$.
(2)$$\begin{array}{cc} \frac{dC_{D}}{dt} & =-\left(\beta +\eta \right)C_{D}, \end{array}$$
(3)$$\begin{array}{cc} \frac{dC_{T}}{dt} & =\eta C_{D}+k\left(t\right)F\left(T\right)C_{T}-\left(\xi +\epsilon +\lambda \right)C_{T}+\theta TC_{M}-\alpha TC_{T}, \end{array}$$
(4)$$\begin{array}{cc} \frac{dC_{M}}{dt} & =\epsilon C_{T}-\theta TC_{M}-\mu C_{M}, \end{array}$$
(5)$$\begin{array}{cc} \frac{dC_{E}}{dt} & =\lambda C_{T}-\delta C_{E}, \end{array}$$
(6)$$\begin{array}{cc} \frac{dT}{dt} & =rT\left(1-bT\right)-\gamma f\left(C_{F},T\right)T, \end{array}$$
in which the total number of functional CAR-T cells is
$$C_{F}=C_{D}+C_{T},$$
and the functions k(t), F(T), and $f(C_{F},T)$ are given by
(7)$$\begin{array}{cc} k(t) & =r_{min}+\frac{p_{1}}{1+{\left(p_{2}\;t\right)}^{p_{3}}}\;, \end{array}$$
(8)$$\begin{array}{cc} F(T) & =\frac{T}{A+T}\;, \end{array}$$
(9)$$\begin{array}{cc} f(C_{F},T) & =\frac{\frac{C_{F}}{T}}{\vartheta +\frac{a+C_{F}}{T}}\;. \end{array}$$

k(t)F(T) is the antigen-specific expansion rate of the effector cells,

The expansion function F(T) describes the growth of effector CAR-T cells, which only happens in the presence of an antigen and is capped by the cells’ inherent growth potential. This is captured in the relation
$$0\leq \frac{T}{A+T}\leq 1\;.$$The term k(t) influences the antigen-dependent growth of CAR-T cells.

This model represents the interaction mechanisms provided in Figure 1.

### 2.2. Parameters of the ODE System

The model parameters, its units, and their biological meaning are displayed in Table 1 and Table 2. These tables from the article [17] are included here to facilitate readability and a clearer understanding of the model dynamics.

The *kinetic parameters* are crucial figures in the study of biological and chemical processes. They describe the rates at which reactions occur, providing insights into how drugs are absorbed, distributed, metabolized, and eliminated from the system, helping to determine an appropriate dosage.

In this article, we analyzed these kinetic parameters, which are crucial for understanding how different treatments affect patients. The distinction between Table 1 and Table 2 indicates that, while some of these parameters vary from patient to patient, others remain constant across the board, as can be found in the literature [14,17,29] (for more details see Section 3.1). The aim of this article was to provide an estimation of the 13 kinetic parameters that are patient-dependent (Table 2).

Traditionally, researchers estimate these parameters using a *Frequentist* approach [17]. However, our study proposes a shift towards a *Bayesian* methodology, specifically employing Markov chain Monte Carlo (MCMC) methods like Metropolis–Hastings. This choice is driven by the complex nature of biological systems, where individual variability plays a significant role and the data for each patient are not extensive.

A Bayesian approach, in contrast to the frequentist method, allows us to incorporate prior knowledge and uncertainty into our estimates. This is particularly advantageous in the fields of biology and medicine, where it enables a better understanding of underlying processes. By better capturing the variability inherent in biological systems, we can tailor treatments more precisely to individual patients.

This shift towards personalized medicine, facilitated by Bayesian methods, represents a significant advancement. By acknowledging and accounting for individual differences, we can develop more effective and specific treatment plans. This not only improves the efficacy of interventions, but also potentially reduces the risk of adverse effects.

### 2.3. Analyses of the ODEs’ Biological Meaning

This model presumes that, following their administration, CAR-T cells are evenly distributed within a patient’s body, leading to a uniform interaction with tumor cells throughout the system.

In Equation (Equation 2), where *distributed CAR-T cells $\left(C_{D}\right)$* are described, the terms are represented by the following biological interactions:$C_{D}$ have a built-in death rate, denoted by $\beta C_{D}$, reflecting the natural death of these cells over time;$C_{D}$ have a rate at which they successfully take hold, or engraft, symbolized by $\eta C_{D}$, which indicates their ability to persist in the patient’s body and begin their therapeutic function.

Then, we obtain the equation $C_{D}\left(t\right)=C_{D}\left(0\right)e^{-(\beta +\eta )t}$, which describes how the number of CAR-T cells changes over time, where $C_{D}\left(0\right)$ is the initial number of cells introduced into the body, *the initial dose*. This equation captures the net effect of the simultaneous processes of cell death and engraftment. Moreover, we can suppose that
$$[C_{T}\left(0\right),C_{M}\left(0\right),C_{E}\left(0\right)]=\left[0,0,0\right],$$

Since when the infusion starts, all the CAR-T cells are “distributed” and have not yet transitioned into other functional states.

In Equation (3), concerning *Effector CAR-T Cells $\left(C_{T}\right)$*, we can observe the following dynamics:$C_{T}$ cells increase by a factor $\eta C_{D}$ due to the transition of distributed cells into active, effector cells;We previously discussed the $k\left(t\right)F\left(T\right)C_{T}$ term, which modulates the interaction between effector cells and tumor cells;$C_{T}$ cells decrease over time due to a natural dying-off process at a rate represented by $\xi C_{T}$, which models the finite lifespan of these cells in the body;$C_{T}$ cells can transform into memory CAR-T cells at a rate of $\epsilon C_{T}$, contributing to the persistence of therapeutic activity even after effector cells decrease;$C_{T}$ cells gradually lose their growth and cancer-fighting abilities, transitioning into exhausted cells at a rate $\lambda C_{T}$, which reflects the eventual decrease in their functional capacity;The $C_{T}$ population is also reinforced by the transformation into memory CAR-T cells, due to ongoing contact with tumor cells. This is represented by the term $\theta TC_{M}$, indicating that memory cells can be reactivated under certain conditions;Furthermore, tumor cells have various ways to suppress effector CAR-T cells, one of which is described by the term $\alpha TC_{T}$, highlighting the immune-evasive properties of the tumor.

In Equation (4), where the behavior of *Memory CAR-T Cells $\left(C_{M}\right)$* is characterized, we can see the following interactions:$C_{M}$ cells increase by a factor $\epsilon C_{T}$ due to the conversion from effector cells, providing a reservoir of cells that can potentially re-engage in fighting cancer;The $C_{M}$ population decreases by a factor $\theta TC_{M}$, accounting for their transformation back into effector CAR-T cells upon contact with tumor cells, thus maintaining a balance between different CAR-T cell states;Finally, $C_{M}$ cells decrease due to their eventual natural death at a rate $\mu C_{M}$.

Memory CAR-T cells generally live longer than active CAR-T cells, hence μ<ξ. It is important to note that the memory form of CAR-T discussed here does not have direct cancer-killing properties. The creation of the memory CAR-T cell pool primarily occurs during the phase where active CAR-T cells decrease in number, serving as a potential backup for future immune responses.

Moving to Equation (5), we see the *exhausted CAR-T cell $\left(C_{E}\right)$* population described, where the following terms are observed:$C_{E}$ cells increase by a factor $\lambda C_{T}$, as effector cells gradually lose their growth and cancer-fighting abilities, transitioning into a less functional state;The natural death of these exhausted cells is accounted for by the term $\delta C_{E}$, representing their eventual elimination from the body.

Finally, in Equation (6), the *tumor cells (T)* are assumed to grow according to a logistic model with a growth rate *r* and a carrying capacity of 1/b, represented by
rT(1−bT).

Moreover, the effectiveness of functional CAR-T cells in eliminating tumor cells is given by $\gamma f(C_{F},T)$. The function $f(C_{F},T)$ indicates the degree of interaction between CAR-T cells and tumor cells, with a saturation point when $T\gg C_{F}$, reflecting a scenario where the tumor burden exceeds the CAR-T cell capacity. If $T\ll C_{F}$, the tumor cell killing rate is approximately γ, implying near-optimal efficacy. The term θT defines the cell number where the function reaches half its maximum effect, representing a non-linear relationship between CAR-T cell activity and the tumor.

Details of the technical analysis regarding the existence and uniqueness of solutions are provided in Appendix A.

### 2.4. MCMC Methods

Markov chain Monte Carlo (MCMC) methods are powerful tools for parameter estimation and used in our Bayesian framework. These methods work by generating samples from a probability distribution based on a defined model and the observed data. The idea is to construct a Markov chain, a sequence of random samples, that has the desired distribution as its equilibrium distribution. Over time, the distribution of these samples converges to the true posterior distribution of the parameters, allowing for accurate estimation, even in complex models.

The core advantage of MCMC methods lies in their ability to incorporate prior knowledge about the parameters into the estimation process. Unlike frequentist methods, which focus solely on maximizing the likelihood function to find “best fit” parameters, Bayesian methods—facilitated by MCMC—integrate both likelihood and prior information. This is particularly beneficial in situations where data are sparse or noisy.

In the context of CAR-T therapy, where patient measurements are often limited due to the invasive and costly nature of the treatment, MCMC methods are a useful tool. The ability to include prior biological knowledge or expert opinion about the expected behavior of the therapy as priors can significantly improve the robustness of the parameter estimates. This is crucial in such settings, because frequentist methods, which rely solely on the available data, may produce a “best solution” that is not accurate when data are sparse. The Bayesian approach, by contrast, allows for more informed and credible inferences about the parameters, leading to better decision-making in clinical contexts.

Next, we will focus on Metropolis–Hastings variants mixed with a differential evolution (DE) approach. For detailed background information on these methods, please refer to the Appendix A.

#### 2.4.1. DEMetropolis Algorithm

The DEMetropolis algorithm is a powerful hybrid method that combines the principles of differential evolution (DE) with the Metropolis–Hastings criterion. This algorithm generates new candidate solutions by mutating differences between randomly selected parameter chains, allowing for efficient exploration of the parameter space. Based on extensive evaluation criteria, including the speed of convergence and the minimization of bias between actual and simulated data, we determined that a combination of Metropolis–Hastings and DE is optimal for our analysis.

As shown in Appendix A, the *differential evolution* (DE) algorithm is a genetic algorithm used for global optimization problems. The DEMetropolis algorithm, derived from DE and integrated with the *Metropolis–Hastings* criterion (Appendix A), generates new candidate vectors by mutating the difference between randomly selected chains. The method, described in [24], uses a single tuning factor, where the acceptance of these new candidates is governed by the Metropolis–Hastings rule, ensuring that the algorithm not only converges quickly but also maintains a balance between exploration and exploitation, making it highly effective for complex stochastic models.

#### 2.4.2. DEMetropolisZ Algorithm

The DEMetropolisZ algorithm represents another step further in parameter estimation accuracy. The method builds upon the DEMetropolis framework by incorporating a unique strategy, the use of information from the previous jumps, in its sampling approach. The modification of the sampling approach modifies the trajectory to be in parallel with the line $\theta_{a}-\theta_{b}$ and is thus designated as a parallel direction update. By utilizing parallel direction sampling [30,31], we adopt Gibbs sampling in this specific direction, an approach that enables local movement. Each update spans a dimension of at least min(d,N−1). Consequently, the sampling strategy must address a space greater than dimension *d* to be effective.

To bypass the condition that N>d, it is possible to use difference vectors from prior updates to improve the chains. Integrating this strategy converts the standard DEMetropolis into an *adaptive MCMC process* [32,33]. To ensure consistency in converging to the posterior distribution, the updates must gradually decrease in magnitude over time [34]. This objective is achievable by uniformly sampling the difference vectors at random, potentially from a version of the historical data that have undergone thinning. The practice of thinning offers several benefits, not least of which is a reduction in the demands on data storage.

### 2.5. PyMC Model

Now, we can specify the PyMC [28] model for the MCMC algorithms. We used a *normal likelihood* on untransformed data (i.e., not log transformed) to best fit the peaks of the data:(10)$$L\left(\theta |x\right)=\frac{1}{\sqrt{2\pi \sigma^{2}}}\exp\left(-\frac{{\left(x-\mu \right)}^{2}}{2\sigma^{2}}\right).$$

Here, *x* represents the actual data points, and μ represents the simulated ODE predictions. The term (x−μ) denotes the error between the actual data and the simulated predictions.

The *priors* are represented by *truncated normal* distributions, defined as
$$p\left(\theta \right)=\frac{\phi (\theta ;\mu ,\sigma )}{\Phi (b;\mu ,\sigma )-\Phi (a;\mu ,\sigma )}$$
for θ in the interval [a,b], where ϕ(θ;μ,σ) is the probability density function of the normal distribution with mean μ (the value obtained from least square) and standard deviation σ (chosen specifically for each parameter), and Φ(·;μ,σ) is the cumulative distribution function of the normal, evaluating the accumulated probability up to the specified point. Here, a=0 and b=+∞ are the lower and upper limits of truncation, respectively. The lower bound is set to ensure that all parameters are positive for biological reasons, which also guarantees the existence of the ODE solution.

## 3. Results

We analyzed the practical aspects of parameter estimation; our focused was to bridge the theoretical foundations with tangible, computational demonstrations, using a suite of algorithms implemented in Python. This approach was facilitated by the use of PyMC [28], a powerful library from Python for probabilistic programming, which enabled us to conduct sophisticated Bayesian inference. The code was run on Oracle VM VirtualBox, a virtual machine for Linux, that allowed us to create more chains in parallel.

We explored four distinct algorithms within the PyMC framework:*Metropolis*—standard Metropolis–Hastings;*DEMetropolis*—a differential evolution Metropolis;*DEMetropolisZ*—a differential evolution Metropolis sampler that uses the past to inform sampling jumps;*SMC*—sequential Monte Carlo.

All of them offer unique advantages for parameter estimation, as shown in Table 3.

These algorithms were chosen for their relevance and efficiency in handling complex statistical problems, providing a comprehensive overview of current methodologies in Bayesian inference.

We compared the performance of these Bayesian algorithms against the classical least squares method using scipy.optimize.least_squares [35], a staple in statistical modeling for parameter estimation. Through the careful examination of these algorithms and their applications, we present a pathway for innovative research directions and the advancement of statistical methodologies in the mathematical sciences.

### 3.1. Setting for CAR-T Cell Therapy

We remind that the model [17] consists of the ODE systems (2)–(9). Moreover, we recall that the state vector $\left[C_{D}\left(t\right),C_{T}\left(t\right),C_{M}\left(t\right),C_{E}\left(t\right)\right]$ indicates the population of the Car-T phenotypes, respectively:Distribution ($C_{D}$);Effector ($C_{T}$);Memory ($C_{M}$);Exhausted ($C_{E}$).

T(t) is the population of the tumor cells.

The initial conditions of the states are
$$\left[C_{D}\left(0\right),C_{T}\left(0\right),C_{M}\left(0\right),C_{E}\left(0\right)\right]=\left[\mathrm{CAR-T}\;\mathrm{cell}\;\mathrm{dose},0,0,0,10^{7}\right].$$

The CAR-T cell dose depends on the patient.

In Table 4, the fixed parameters values taken from the literature are shown.

The aim of this study was to give an estimation of the 13 kinetic parameters:$$\left[\beta ,\eta ,r_{\min},p_{1},p_{2},p_{3},A,x_{i},\epsilon ,\lambda ,\mu ,\delta ,\gamma \right].$$

These are the unknowns that we wished to infer from experimental data.

In Table 5, the medical data of a specific patient (Patient 28) are shown from a study performed in 2021 [15].

Note that, in the studies, only the total number of Car-T cells was shown:(11)$$C=C_{D}+C_{T}+C_{M}+C_{E},$$
without considering the different phenotypes, since it was not possible to differentiate them. In Figure 2, we show the plot of *C* (Equation 11), which is in line with the results of A. Paixao. et al. [17].

### 3.2. Least Squares Results

The least_squares [35] function was used to perform non-linear least squares optimization. The algorithm can be summarized as follows:We define the models (2)–(9);We compare the medical data in Table 5 and the model numerical solution obtained with Runge–Kutta [36], using the solver odeint [35];We set the initial guess for parameters using parameters that the authors obtained in the article [17], shown in Table 6;We used the least_squares function with the bound of having positive parameters, to ensure the existence of the ODEs;The procedure returned the values of the best parameters. We could compare these with the parameters chosen in the article, as shown in Table 7.

In Figure 3, we can observe the plot obtained using the parameters of the article [17] compared with the plot obtained using least_squares solutions. The plot obtained with the least squares method was a better fit compared to the one in the previous article [17]. This improvement in fit indicated that our model captured the underlying trend in the data more accurately.

However, as we pointed out in Section 2.4, we aimed to perform a Bayesian estimation for several reasons. Firstly, our dataset was relatively small, and Bayesian methods are well-suited for situations with limited data. Secondly, they allowed us to incorporate prior knowledge and quantify the uncertainty in our estimates, providing a more comprehensive understanding of the parameter distributions. This approach can give us a general idea of the underlying patterns and potential variability in the data, which is crucial for drawing more informed and robust conclusions.

### 3.3. MCMC Results

The results of utilizing MCMC methods are presented in Table 8, which shows the mean and standard deviation of the *posterior* distribution for each parameter. We also give an estimation of σ, which is the variance in the Likelihood function (Equation 10).

From the values in Table 8, we obtained the trace plots of the four MCMC algorithms, represented in Figure 4. The trace plots show that we obtained a highly successful Bayesian estimation process, demonstrating excellent convergence and mixing for all parameters. In fact, each chain showed stable mean values and overlapped well with the others, reflecting an efficient and thorough exploration of the parameter space. The posterior distributions were well-defined and smooth, suggesting a strong model fit to the data. The consistency and clarity of these results highlight the robustness and reliability of our Bayesian estimation approach in accurately capturing the underlying parameter distributions. This comprehensive analysis provided a solid foundation for drawing meaningful conclusions from our limited dataset, ensuring that our parameter estimates were both precise and credible.

Moreover, Figure 5 shows four plots of the total CAR-T cell, each representing a comparison of the medical data Table 5 and inference model runs using different sampling methods. The plots demonstrate that each of the four sampling methods (SMC, Metropolis, DEMetropolis, DEMetropolisZ) provided a good fit to the data, capturing the dynamics of the cell count over the period.

To compare the marginal probability of each parameter for each method, we used plot_forest from the Python library ArviZ [37]. Figure 6 shows plots for every parameter.

These plots showcase a comparison of the marginal probability distributions for a set of parameters estimated using different statistical methods. Marginal probabilities are the probabilities of observing a particular value for a parameter, regardless of the values of other parameters. Essentially, they provide insights into the uncertainty and the possible values each parameter could take after considering all the data and prior information available.

The best method, in this case, was DEMetropolisZ, as it generally showed more peaked and distinct distributions for the parameters. This implies a higher level of confidence in the parameter estimates, as there was less uncertainty associated with the parameter values.

Moreover, the distributions resulting from DEMetropolisZ were smoother and more symmetric, which was an indication of the better convergence properties of the algorithm in the parameter space. This suggests that DEMetropolisZ was more efficient in exploring the parameter space and finding the regions of higher probability, which is an important aspect of a good Markov chain Monte Carlo (MCMC) method.

The Metropolis algorithm seemed to produce wider distributions, indicating more uncertainty and less precision in the parameter estimates. On the other hand, SMC appeared to give intermediate results, with varying degrees of uncertainty across the different parameters.

To conclude, DEMetropolisZ stood out as the best method for estimating the marginal probabilities of these parameters, due to its more precise and confident distributions, which indicated a better performance in sampling from the posterior distribution. This could also be observed in the computational times, as we can see in Table 9.

A pair plot is a visual representation of the posterior correlations between various parameters in a statistical model, as obtained by the DEMetropolisZ algorithm. It is a powerful tool for understanding the relationship between multiple variables after Bayesian inference has been performed.

In Figure 7, in which a pair plot of the posteriors correlations is shown, each row and column corresponds to a different parameter, with the histograms on the diagonal showing the marginal distributions of each parameter. The off-diagonal plots show scatter plots for pairs of parameters, which allow us to visually inspect their correlations.

Overall, this pair plot could help in diagnosing potential issues with the statistical model, like identifying parameters that are perhaps too tightly linked or by suggesting transformations or model re-parameterizations to improve the independence of the parameters, leading to more efficient sampling by the MCMC algorithm for future simulations.

Let us look more deeply into the specific parameters in Figure 7.

The histograms (diagonal) exhibit the shapes of the marginal posterior distributions for each parameter. The sharpness of the peaks indicates where the data suggest the most credible values lie;The scatter plots (off-diagonal) indicate the degree and pattern of correlation between pairs of parameters. For instance, a circular cloud of points suggests little to no correlation, whereas an elliptical shape oriented along a line indicates a stronger correlation;Some parameters appear to have little correlation with others, as indicated by the round shapes of their scatter plots. This suggests that these parameters independently contributed to the model;If any of the scatter plots showed a very narrow, elongated ellipse, that would indicate a high degree of correlation, implying that one parameter could be predicted from the other. However, in this plot, while some mild correlations are visible, there does not appear to be an excessively strong linear relationship between any pair of parameters;The distribution shapes and scatter plot orientations provide insight into the potential complexity of the statistical model and the interactions between parameters;The marginal distributions (diagonal plots) for parameters like η and σ are fairly symmetrical and bell-shaped, which suggests a well-defined mean value and suggests that the DEMetropolisZ algorithm did a good job of exploring the parameter space around the peak probability;For parameters like p1, p2, and p3, we see distributions that are slightly skewed, which could indicate that the underlying data have some asymmetries or that these parameters are not normally distributed within the model context. This makes sense, since they are part of the same non-linear function in the ODE system (Equation 7):
(12)$$k\left(t\right)=r_{min}+\frac{p_{1}}{1+{\left(p_{2}\;t\right)}^{p_{3}}};$$Scatter plots for pairs like (β,γ) and (γ,η) show some degree of correlation, as indicated by the elliptical shapes. This is consistent with the model, since these parameters describe the same CAR-T phenotypes, distributed ($C_{D}$). Moreover, these correlations do not appear to be very strong, which is good because this means that the parameters are relatively independent of each other, and the model did not suffer from multicollinearity issues;Some plots, like those involving $r_{\min}$, show a tighter clustering of points, suggesting a strong correlation or interdependence between $r_{\min}$ and other parameters such as beta, delta, and epsilon. This might be important for understanding how changes in $r_{\min}$ affect the model or vice versa.

We can conclude that the scatter plots indicate that the algorithm efficiently explored the parameter space, as evidenced by the spread of points without obvious patterns of constraint or limitation, which would suggest areas of the space not being sampled. Moreover, the marginal distributions are smooth and unimodal, which often indicates good convergence. Sharp peaks in the distributions suggest that the algorithm effectively zeroed in on the most likely parameter values. Even in cases where parameters exhibit correlations, DEMetropolis seemed to handle these well, as indicated by the lack of extreme narrow “banana shapes” in the scatter plots, which would suggest difficulty sampling from correlated distributions.

## 4. Discussion

This work has presented a comprehensive mathematical framework for modeling CAR-T therapy, a groundbreaking approach in the treatment of certain types of cancer. We explored the intricacies of ordinary ODEs in representing the complex dynamics of CAR-T cell phenotypes and their interactions within the human body.

A significant emphasis was placed on advocating for a Bayesian approach over the traditional frequentist methods, such as the least squares optimization algorithm. The Bayesian paradigm allows for the integration of prior medical knowledge and the uncertainties inherent in biological systems, which is critical in the context of personalized medicine. Through the use of Markov chain Monte Carlo (MCMC) methods, we were able to demonstrate how theoretical mathematical constructs can be effectively applied to estimate parameters that are otherwise challenging to ascertain. MCMC methods, supported by the theoretical foundation, enable the selection of suitable priors and likelihoods that are tailored to each unique clinical situation, offering a more robust and individualized model of CAR-T therapy. In this paper, we used standard priors and likelihoods, but in the future, they could be chosen to represent reality in the best way.

Future work could investigate improving these computational models by incorporating machine learning and neural networks. The integration with advanced computing architectures, particularly those utilizing GPU capabilities from providers like NVIDIA, promises to substantially accelerate the computational process. This acceleration could potentially enable real-time customization of therapeutic strategies, making the clinical application of CAR-T cell therapy not only more effective but also more efficient.

The Bayesian approach, therefore, is not just a theoretical preference; it represents a paradigm shift towards a more nuanced and precise understanding of complex medical treatments. As the field of CAR-T cell therapy evolves, the incorporation of such advanced computational techniques will undoubtedly play a fundamental role in the realization of truly personalized medicine. When a patient qualifies for CAR-T therapy, they can be assigned to a specific cohort based on biometric and demographic characteristics such as age, gender, and BMI. Leveraging historical data accumulated for each cohort, we can estimate the kinetic parameters relevant to that group. By identifying the patient’s cohort and applying these kinetic parameters, we can predict the patient’s overall CAR-T cell expansion response using forward ODE simulations. This approach could enable the administration of the optimal dosage to achieve the desired levels of effector and memory CAR-T cells.

## 5. Conclusions

We developed a comprehensive mathematical framework using ordinary differential equations (ODEs) and Bayesian parameter estimation to model CAR-T cell dynamics. This approach enables a deeper understanding of the behavior and interactions of CAR-T cells within the human body, providing insights into their proliferation, memory, exhaustion, and cytotoxic activities. By employing advanced MCMC methods such as Metropolis–Hastings, DEMetropolis, and DEMetropolisZ, we demonstrated how Bayesian techniques offer significant advantages over traditional frequentist methods, particularly in dealing with the complexities of biological systems and individual variability.

Our results underscore the potential of Bayesian inference to improve the predictive accuracy and clinical efficacy of CAR-T cell therapy. The ability to incorporate prior knowledge and quantify uncertainty allows for a more tailored approach to treatment, aligning with the goals of personalized medicine. By providing valuable insights into the proliferation and cytotoxic activity of CAR-T cells, this approach contributes to ongoing efforts to overcome challenges associated with this innovative treatment modality. In the future, this model could be improved using machine learning and neural networking. Using real clinical data, we could classify and offer a specific treatment based on the patient’s characteristics. Opening the door to personalize medicine in CAR-T.

## Figures and Tables

**Figure 2 cancers-16-03132-f002:**
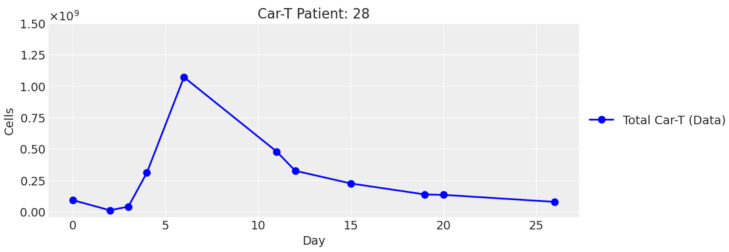
Plot of medical data for patient 28.

**Figure 3 cancers-16-03132-f003:**
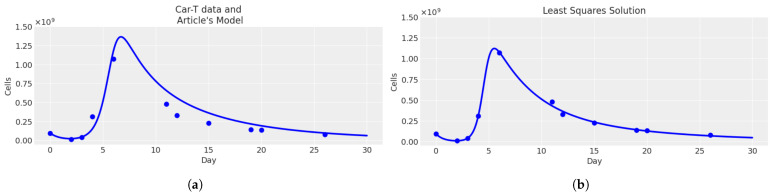
Comparison between the plots of the total CAR-T cells (**—**) obtained with the parameters from the article (**a**) and the least_squares method (**b**). The medical data (•) from Table 5.

**Figure 4 cancers-16-03132-f004:**
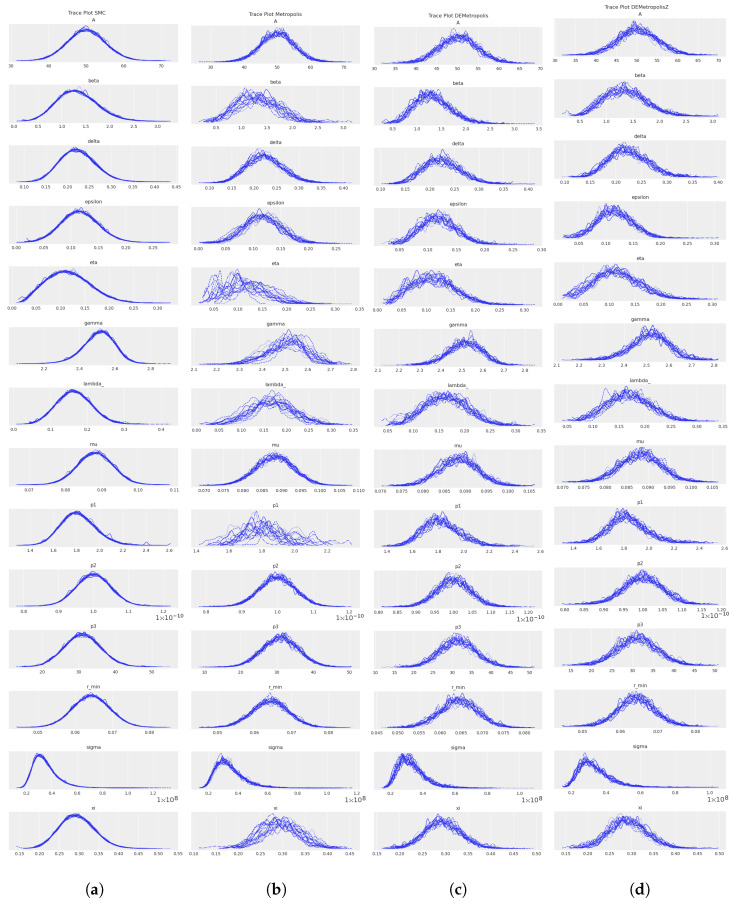
Comparison between the trace plots of kinetic parameters obtained with SMC (**a**), Metropolis (**b**), DEMetropolis (**c**), and DEMetropolisZ (**d**). Each plot was obtained with 10,000 draws and 16 chains in parallel.

**Figure 5 cancers-16-03132-f005:**
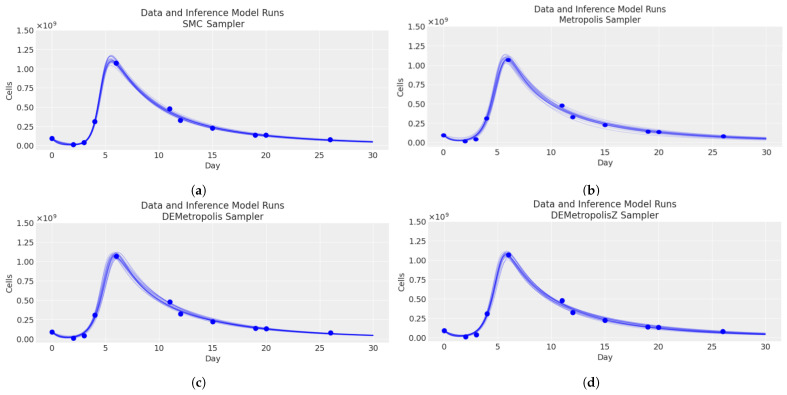
Comparison between the total CAR-T cell plots of the different trajectories for each of the algorithms (—). SMC (**a**), Metropolis (**b**), DEMetropolis (**c**), and DEMetropolisZ (**d**) and the medical data (•) from Table 5.

**Figure 6 cancers-16-03132-f006:**
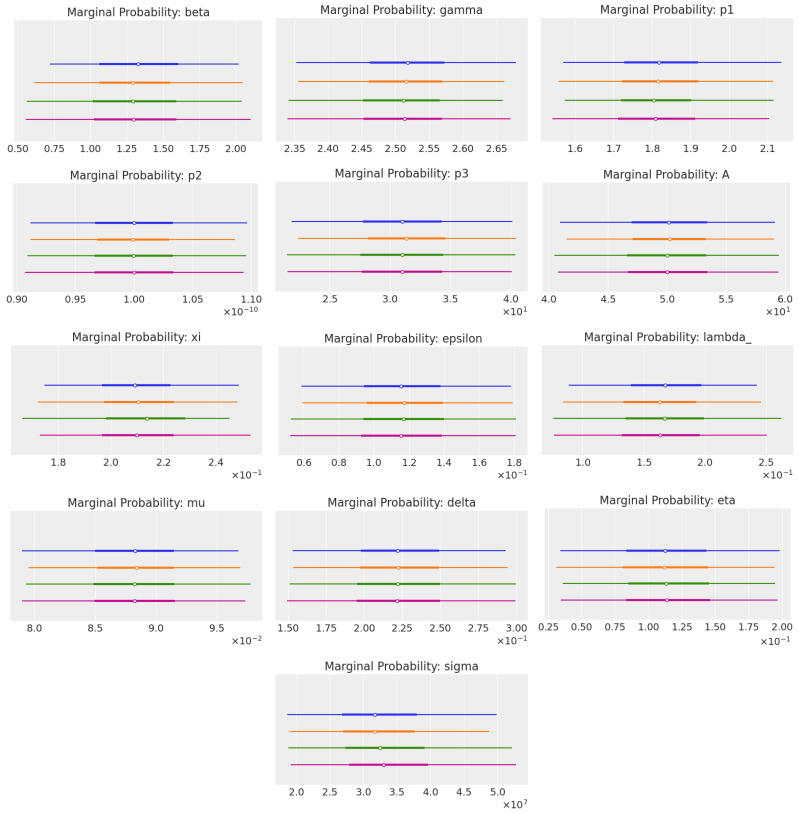
Forest plots of marginal probabilities (“dispersion coefficient” = (97.5th percentile value − 2.5th percentile value)/mean value) of every kinetic parameter. These plots compare four methods: DEMetropolisZ (**—**), DEMetropolis (**—**), Metropolis algorithm (**—**), and sequential Monte Carlo, SMC (**—**).

**Figure 7 cancers-16-03132-f007:**
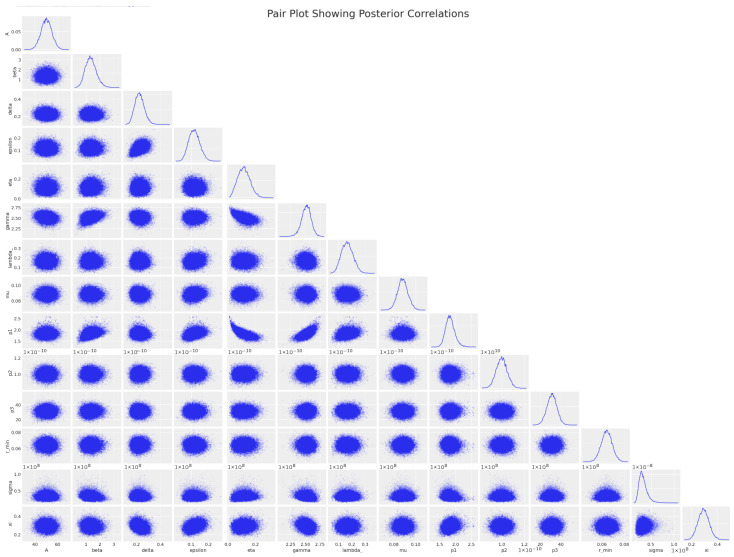
DEMetropolisZ pair plot of the posterior correlations.

**Table 1 cancers-16-03132-t001:** Non-patient-dependent parameters with their units and biological meanings.

Parameter	Unit	Biological Meaning
α	(cell·day) ^−1^	Inhibition coefficient of effector CAR-T cells due to interaction with tumor cells.
*r*	day ^−1^	Maximum growth rate of tumor cells.
*b*	cell ^−1^	Inverse of the carrying capacity of tumor cells.
ϑ	-	Half-saturation constant of the cytotoxic effect on tumor cells.
θ	(cell·day) ^−1^	Conversion coefficient of memory CAR-T cells into effector CAR-T cells due to interaction with tumor cells.
*a*	cell	Half-saturation constant of $f(C_{T},T)$.

**Table 2 cancers-16-03132-t002:** Patient parameters with their units and biological meanings.

Parameter	Unit	Biological Meaning
β	day ^−1^	Reduction rate of infused cells due to natural death during their distribution.
η	day ^−1^	Engraftment rate of injected cells to blood and tumor niche.
$r_{\min}$	day ^−1^	Minimum expansion rate of effector CAR-T cells.
$p_{1}$	day ^−1^	Initial expansion rate of effector CAR-T cells.
$p_{2}$	day ^−1^	Rate that regulates the duration of maximum expansion period of effector CAR-T cells.
$p_{3}$	-	Expansion coefficient that regulates decay of maximum expansion period of effector CAR-T cells.
*A*	cell	Half-saturation constant of F(T).
ξ	day ^−1^	Death rate of effector CAR-T cells.
ϵ	day ^−1^	Conversion rate of effector CAR-T cells into memory CAR-T cells.
λ	day ^−1^	Exhaustion rate of effector CAR-T cells.
μ	day ^−1^	Death rate of memory CAR-T cells.
δ	day ^−1^	Death rate of exhausted CAR-T cells.
γ	day ^−1^	Cytotoxic rate of functional CAR-T cells on tumor cells.

**Table 3 cancers-16-03132-t003:** MCMC algorithms with their advantages and disadvantages.

Algorithm	Advantages	Disadvantages
Metropolis	Simple to implement; Works well for low-dimensional problems; Requires fewer assumptions about the target distribution.	Can be slow to converge; Sensitive to the choice of proposal distribution; Inefficient for high-dimensional or complex posterior distributions.
DEMetropolis	Adapts to the posterior distribution; Can handle high-dimensional and multimodal distributions.	Can be computationally expensive; Requires a large population size for effective sampling.
DEMetropolisZ	Better mixing and convergence properties compared to DEMetropolis; More efficient posterior distribution adaptation.	Requires careful tuning of parameters’ prior distributions; Can be computationally intensive.
SMC	Can handle multimodal distributions; Effective for complex posterior landscapes; Adaptable to changing target distributions.	Slow convergence and computationally intensive; Requires careful tuning of resampling steps; Less efficient for high-dimensional problems.

**Table 4 cancers-16-03132-t004:** Fixed parameters, including their respective units, values, and references to the articles from which they were taken.

Parameter	Unit	Value	Reference
α	(cell·day) ^−1^	5.500 × 10^−7^	[17]
*r*	day ^−1^	1.760 × 10^−1^	[29]
*b*	cell ^−1^	5.000 × 10^−13^	[29]
ϑ	-	3.050 × 10^−1^	[29]
θ	(cell·day) ^−1^	6.000 × 10^−6^	[14]
*a*	cell	1.000 × 10^3^	[17]

**Table 5 cancers-16-03132-t005:** Medical data of patient 28 [15]. Total number of CAR-T cells (C) with the corresponding day.

Day	Total CAR-T Cells (C)
0	9.230 × 10^7^
2	1.128 × 10^7^
3	4.029 × 10^7^
4	3.106 × 10^8^
6	1.070 × 10^9^
11	4.786 × 10^8^
12	3.259 × 10^8^
15	2.245 × 10^8^
19	1.372 × 10^8^
20	1.340 × 10^8^
26	7.801 × 10^7^

**Table 6 cancers-16-03132-t006:** Kinetic parameters value for Patient 28 taken from article [15].

Parameter	Value
β	1.051
η	5.400 × 10^−2^
$r_{\min}$	1.000 × 10^−3^
$p_{1}$	1.750
$p_{2}$	7.539 × 10^−25^
$p_{3}$	3.100 × 10^1^
*A*	5.000 × 10^1^
ξ	2.548 × 10^−1^
ϵ	6.000 × 10^−2^
λ	1.000 × 10^−1^
μ	9.010 × 10^−2^
δ	1.498 × 10^−1^
γ	2.250

**Table 7 cancers-16-03132-t007:** Comparison of the parameter values taken from article [17] and obtained with the least squares function.

Parameter	Article Value	Least Squares Solution	Difference
β	1.051	1.412	3.611 × 10^−1^
η	5.400 × 10^−2^	6.432 × 10^−3^	4.757 × 10^−2^
$r_{\min}$	1.000 × 10^−3^	4.127 × 10^−1^	4.117 × 10^−1^
$p_{1}$	1.750	2.116	3.657 × 10^−1^
$p_{2}$	7.539 × 10^−25^	1.000 × 10^−10^	1.000 × 10^−10^
$p_{3}$	3.100 × 10^1^	3.100 × 10^1^	0
*A*	5.00 × 10^1^	5.159 × 10^1^	1.591
ξ	2.548 × 10^−1^	2.121 × 10^−1^	4.263 × 10^−2^
ϵ	6.000 × 10^−2^	8.849 × 10^−2^	2.849 × 10^−2^
λ	1.000 × 10^−1^	7.111 × 10^−2^	2.889 × 10^−2^
μ	9.010 × 10^−2^	9.036 × 10^−2^	2.558 × 10^−4^
δ	1.498 × 10^−1^	3.121 × 10^−1^	1.624 × 10^−1^
γ	2.250	2.796	5.465 × 10^−1^

**Table 8 cancers-16-03132-t008:** Means and standard deviations of the posterior distribution of the parameters with MCMC methods.

Parameter	SMC			Metropolis		
	Mean	SD		Mean	SD	
β	1.318	4.160 × 10^−1^		1.312	4.070 × 10^−1^	
η	1.170 × 10^−1^	4.500 × 10^−2^		1.160 × 10^−1^	4.400 × 10^−2^	
$r_{\min}$	6.400 × 10^−2^	5.000 × 10^−3^		6.400 × 10^−2^	5.000 × 10^−3^	
$p_{1}$	1.816	1.510 × 10^−1^		1.818	1.430 × 10^−1^	
$p_{2}$	1.010 × 10^−10^	1.400 × 10^−11^		1.012 × 10^−10^	1.020 × 10^−11^	
$p_{3}$	3.105 × 10^1^	4.939		3.099 × 10^1^	5.032	
*A*	5.007 × 10^1^	4.962		5.001 × 10^1^	5.043	
ξ	2.950 × 10^−1^	4.400 × 10^−2^		2.960 × 10^−1^	4.400 × 10^−2^	
δ	2.240 × 10^−1^	4.000 × 10^−2^		2.240 × 10^−1^	4.000 × 10^−2^	
ϵ	1.160 × 10^−1^	3.400 × 10^−2^		1.170 × 10^−1^	3.400 × 10^−2^	
λ	1.650 × 10^−1^	4.700 × 10^−2^		1.670 × 10^−1^	4.900 × 10^−2^	
μ	8.800 × 10^−2^	5.000 × 10^−3^		8.800 × 10^−2^	5.000 × 10^−3^	
γ	2.508	8.800 × 10^−2^		2.507	8.500 × 10^−2^	
σ	3.476 × 10^7^	9.728 × 10^6^		3.421 × 10^7^	9.683 × 10^6^	
**Parameter**	**DEMetropolis**			**DEMetropolisZ**		
	**Mean**	**SD**		**Mean**	**SD**	
β	1.318	3.640 × 10^−1^		1.346	3.960 × 10^−1^	
η	1.140 × 10^−1^	4.500 × 10^−2^		1.150 × 10^−1^	4.400 × 10^−2^	
$r_{\min}$	6.400 × 10^−2^	5.000 × 10^−3^		6.400 × 10^−2^	5.000 × 10^−3^	
$p_{1}$	1.827	1.510 × 10^−1^		1.829	1.510 × 10^−1^	
$p_{2}$	1.010 × 10^−10^	1.030 × 10^−11^		1.013 × 10^−10^	1.010 × 10^−11^	
$p_{3}$	3.141 × 10^1^	4.791		3.105 × 10^1^	4.847	
*A*	5.025 × 10^1^	4.655		5.024 × 10^1^	4.807	
ξ	2.950 × 10^−1^	4.000 × 10^−2^		2.920 × 10^−1^	4.300 × 10^−2^	
δ	2.240 × 10^−1^	3.800 × 10^−2^		2.240 × 10^−1^	3.800 × 10^−2^	
ϵ	1.180 × 10^−1^	3.200 × 10^−2^		1.170 × 10^−1^	3.200 × 10^−2^	
λ	1.630 × 10^−1^	4.300 × 10^−2^		1.690 × 10^−1^	4.100 × 10^−2^	
μ	8.800 × 10^−2^	5.000 × 10^−3^		8.800 × 10^−2^	5.000 × 10^−3^	
γ	2.513	8.200 × 10^−2^		2.515	8.600 × 10^−2^	
σ	3.107 × 10^7^	8.603 × 10^6^		3.331 × 10^7^	9.154 × 10^6^	

**Table 9 cancers-16-03132-t009:** Computational times of the algorithms.

Algorithm	Computational Time (s)
SMC	1.675 × 10^4^
Metropolis-Hastings	3.798 × 10^3^
DEMetropolis	6.570 × 10^2^
DEMetropolisZ	3.100 × 10^2^
least_squares	5.910

## Data Availability

Source data are provided in this paper and all data used in this study can be downloaded from the cited sources. All information, such as parameter values, to replicate the simulations and analysis is available in this work.

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
