# Peer review of "MCMC Methods for Parameter Estimation in ODE Systems for CAR-T Cell Cancer Therapy"

_cancers, 2024, doi:10.3390/cancers16183132_

Round 1

Reviewer 1 Report

Comments and Suggestions for Authors

Superb paper very novel and innovative, addressing CAR-T cell dynamics, based on their proliferation and cytotoxic activity using real life patients' samples, establishing a mathematic algorithmic model addressing the key question in the field of CART cells. Hopefully this sophisticated but accurate model will help to improve further CART cell therapeutic yield. Specifically, the authors developed mathematical framework using ordinary differential equations (ODEs) with Bayesian parameter estimation. Bayesian statistics are used to estimate model parameters through Monte Carlo integration, Bayesian inference, and Markov Chain Monte Carlo (MCMC) methods. The paper explores MCMC methods, including the Metropolis-Hastings algorithm and DEMetropolis and  DEMetropolisZ algorithms, which integrate Differential Evolution to enhance convergence rates. 10 Theoretical findings and algorithms are validated using Python and Jupyter Notebooks.

Author Response

Comment 1: Superb paper very novel and innovative, addressing CAR-T cell dynamics, based on their proliferation and cytotoxic activity using real life patients' samples, establishing a mathematic algorithmic model addressing the key question in the field of CART cells. Hopefully this sophisticated but accurate model will help to improve further CART cell therapeutic yield. Specifically, the authors developed mathematical framework using ordinary differential equations (ODEs) with Bayesian parameter estimation. Bayesian statistics are used to estimate model parameters through Monte Carlo integration, Bayesian inference, and Markov Chain Monte Carlo (MCMC) methods. The paper explores MCMC methods, including the Metropolis-Hastings algorithm and DEMetropolis and  DEMetropolisZ algorithms, which integrate Differential Evolution to enhance convergence rates. 10 Theoretical findings and algorithms are validated using Python and Jupyter Notebooks.

Answer 1: Thank you very much for your kind and encouraging feedback. We are thrilled that you found our work on modeling CAR-T cell dynamics to be novel and innovative. We hope that our approach, combining ODEs with advanced Bayesian methods, will indeed contribute to further advancements in CAR-T cell therapies. We greatly appreciate your thoughtful review.

Reviewer 2 Report

Comments and Suggestions for Authors

The authors used statistics method to estimate CAR-T therapy and concluded that MCMC methods for parameter in ODEs system can be used in prediction of CAR-T therapy.

1. Though interesting, could authors provide the validation of the system in real clinic use?

Author Response

Comment 1: Though interesting, could authors provide the validation of the system in real clinic use?

Response 1: Thank you very much for your valuable feedback. We appreciate your interest in our work. As you noted, our current study focuses on the theoretical application of MCMC methods for parameter estimation in ODE systems related to CAR-T therapy. While we have not yet applied this method in a real clinical setting, we recognize the importance of such validation. We have addressed this in our revised manuscript by highlighting that applying our approach in a clinical study will be a critical next step, as mentioned in the conclusion. (pag. 18, Conclusion, rows 468-470)

We look forward to pursuing this in future research.

Thank you again for your constructive comments.

Reviewer 3 Report

Comments and Suggestions for Authors

The authors have done nice job by developing a model to capture the key dynamics of CAR-T cell therapy.

The authors can work on following comments to make the paper comprehensive.

1. Reduce the analogy(plagiarism) to less than 15%.

2. Improve the English language and scientific language of the manuscript.

3. Add more latest references.

4. Add more details in introduction and results to make the formulas clear to the readers.

Comments on the Quality of English Language

Improve the English language and scientific language of the manuscript.

Author Response

Comment 1: Reduce the analogy (plagiarism) to less than 15%.

Response 1: Thank you for pointing this out. We agree with this comment. Therefore, we have replaced the sections (2.2-2.3-2.4.2.5 sections, lines 118-242) with analogies by providing brief summaries instead.

Comment 2: Improve the English language and scientific language of the manuscript.

Response 2: Agree. We have, accordingly, revised the manuscript to improve both the English language and the scientific terminology used.

Comment 3: Add more latest references.

Response 3: Thank you for this suggestion. We have added several recent references in the introduction, specifically related to the use of ODE models and MCMC methods in biology. (pag. 4, Introduction, lines 56-89)

Comment 4: Add more details in the introduction and results to make the formulas clear to the readers.

Response 4: Agree. We have, accordingly, expanded the introduction, (pag. 4, Introduction, lines 56-89).

We have also elaborated further on the formulas in Section 2.3, lines 146-211 lines to ensure clarity for the readers.

Reviewer 4 Report

Comments and Suggestions for Authors

The authors consider a previously-published ODE model for CAR-T cell therapy, and use various MCMC methods to estimate model parameters. 

There are a few issues that need to be addressed before the manuscript can be published:

1) The main objective of this work is not very clear: am I wrong, or the main goal is to compare various algorithms for parameter estimation, and for this the authors use a model they published in [19]?

If this is goal, then the authors need to review the state-of-the-art of the literature focused on parameter estimation (as there are more and more studies focused on Bayesian approaches). I do not think that the few references they mentioned in the Introduction are enough to give the reader the current state of the art of the field. 

This and next comment are particularly relevant, especially give that the authors state at the beginning of the Discussion that "

  1. This work has presented a comprehensive mathematical framework for modeling 406 CAR-T therapy

"

2) The model is poorly described. The authors should add biological references to support the various assumptions incorporated into model equations. So that the reader can understand that the model is biologically realistic (otherwise, anyone can through in some equations, with no connection to biology, and claim that their model is correct biologically). So all section 2.1 should be full of biological references. Same for sections 2.2 and 2.3 !!

3) Why the parameters in Table 1 are patient-dependent, while those in Table 2 are not patient dependent? Why, for example, the conversion coefficient of memory Car-T cells into effector Car-T cells is not patient specific? Biological references need to be added, to back up the various assumptions made here.

The results of this study need to be discussed also with respect to the parameters that the authors chose to keep fixed.

Author Response

Comment 1: The main objective of this work is not very clear: am I wrong, or the main goal is to compare various algorithms for parameter estimation, and for this the authors use a model they published in [19]?

If this is the goal, then the authors need to review the state-of-the-art of the literature focused on parameter estimation (as there are more and more studies focused on Bayesian approaches). I do not think that the few references they mentioned in the Introduction are enough to give the reader the current state of the art of the field.

Response 1: Thank you for pointing this out. We agree with this comment. Therefore, we have added references in the introduction related to recent studies that utilize ODE models and MCMC methods in biology to provide a clearer view of the state-of-the-art. (pag. 4, Introduction, lines 56-89)

Comment 2: The model is poorly described. The authors should add biological references to support the various assumptions incorporated into model equations so that the reader can understand that the model is biologically realistic. So all section 2.1 should be full of biological references. Same for sections 2.2 and 2.3!!

Response 2: Agree. We have, accordingly, expanded Section 2.2 to better explain the biological dynamics described by the model. In addition, we have included more biological references throughout Sections 2.1, 2.2, and 2.3  (lines 118-242) to ensure that the model is well-supported by existing biological studies.

Comment 3: Why are the parameters in Table 1 patient-dependent, while those in Table 2 are not patient-dependent? Why, for example, is the conversion coefficient of memory CAR-T cells into effector CAR-T cells not patient-specific? Biological references need to be added, to back up the various assumptions made here.

Response 3: Thank you for this observation. We have added references explaining why certain parameters are fixed while others are patient-dependent. These references are now connected to the explanations provided in Table 4. (lines 126-131,pag.5 and line 289, pag. 9)

Round 2

Reviewer 4 Report

Comments and Suggestions for Authors

The authors have addressed some of the previous issues but not all:

- pages 1-6 describe mathematical model introduced in ref 17. OK. Now this part is clear.

- Then, on page 7, the authors start presenting various MCMC algorithms; but the presentation is not logical. At the beginning of Section 2.4 (DEMetropolis algorithms) they state "In this section we explain why we chose  specific MCMC methods... Therefore we implemented the DEMetropolis and DEMetropolisW algorithms." Then Section 2.5 discusses the DEMetropolisZ algorithms . Should't this be a sub-section in the previous section? Shouldn't all subsequent sections be sub-sections of section 2.4, which should have some general sentences about MCMC algorithms?

Please present all these algorithms in a logical way, so that the reader can understand the flow of ideas and how the different algorithms are connected with each other

- Figure 3(b) shows an almost perfect fit of the data using least square approaches. The authors have not explained very clearly why is it better to use Bayesian approaches. Is it really better, or does it provide additional information? Does one need to always use Bayesian approaches (and not frequentist approaches), or maybe it depends on the question that one wants to address. 

Please expand this discussion in the context of your results with frequentist and Bayesian approaches. 

Author Response

Comment 1: 

hen, on page 7, the authors start presenting various MCMC algorithms; but the presentation is not logical. At the beginning of Section 2.4 (DEMetropolis algorithms) they state "In this section we explain why we chose  specific MCMC methods... Therefore we implemented the DEMetropolis and DEMetropolisW algorithms." Then Section 2.5 discusses the DEMetropolisZ algorithms . Should't this be a sub-section in the previous section? Shouldn't all subsequent sections be sub-sections of section 2.4, which should have some general sentences about MCMC algorithms?

Please present all these algorithms in a logical way, so that the reader can understand the flow of ideas and how the different algorithms are connected with each other

Response 1: Thank you for pointing this out. We agree with this comment. Therefore, we have have merged sections 2.5 and 2.6 into the new section 2.5. Additionally, we have expanded section 2.4 by providing a brief explanation of MCMC methods. (lines 212-268) 

Comment 2: Figure 3(b) shows an almost perfect fit of the data using least square approaches. The authors have not explained very clearly why is it better to use Bayesian approaches. Is it really better, or does it provide additional information? Does one need to always use Bayesian approaches (and not frequentist approaches), or maybe it depends on the question that one wants to address. 

Response 2: Agree. We have expanded section 2.4 by explaining why we believe a Bayesian approach is more appropriate than a frequentist one in this case. We also added a reference to this near Figure 3b, pointing to section 2.4. (lines 212-268 and line 341)

Round 3

Reviewer 4 Report

Comments and Suggestions for Authors

The manuscript can be accepted for publication